# Whole-Body Neural Policy for Zero-Shot Cross-Embodiment Motion Planning

Prabin Kumar Rath
Arizona State University
prath4@asu.edu

Nakul Gopalan
Arizona State University
ng@asu.edu

*Abstract*—**Classical manipulator motion planners work across different robot embodiments [34]. However they plan on a pre-specified static environment representation, and are not scalable to unseen dynamic environments. Neural Motion Planners (NMPs) [22] are an appealing alternative to conventional planners as they incorporate different environmental constraints to learn motion policies directly from raw sensor observations. Contemporary state-of-the-art NMPs can successfully plan across different environments [9]. However none of the existing NMPs generalize across robot embodiments. In this paper we propose Cross-Embodiment Motion Policy (XMoP), a neural policy for learning to plan over a distribution of manipulators. XMoP implicitly learns to satisfy kinematic constraints for a distribution of robots and *zero-shot* transfers the planning behavior to unseen robotic manipulators within this distribution. We trained XMoP on planning demonstrations from over three million procedurally sampled robotic manipulators in different simulated environments. Despite being completely trained on synthetic embodiments and environments, our policy exhibits strong sim-to-real generalization across manipulators with different kinematic variations and degrees of freedom with the same set of frozen policy parameters. We show sim-to-real demonstrations on two unseen manipulators solving novel planning problems in different real-world environments even with dynamic obstacles. Videos are available at https://sites.google.com/view/xmop.**

## I. Introduction

Motion planning for robotic manipulators is the task of finding a sequence of robot configurations connecting a start joint state to the goal joint state while respecting joint limits of the robot and avoiding obstacles. Even after decades of research in this domain, real-time motion planning in complex unseen environments is still a challenging problem [29, 21, 24].

Classical motion planners either use random sampling to explore the configuration-space [18, 17, 16, 28] or employ gradient-based optimization methods [23, 15, 26, 8] to search for a valid plan. While these algorithms generalize across embodiments, they often demand a non real-time computation budget for generating desired motion behaviors in geometrically complex environments [21, 28]. Furthermore, these algorithms conventionally assume the availability of a pre-computed geometric representation of the robot's workspace for state validation, thus making them unscalable in unseen environments with novel types of obstacles. To overcome these limitations, Neural Motion Planners (NMP) learn to generate trajectories directly from visual observations [22, 12, 14, 33, 5, 9]. However, these policies are individually trained on data from a single manipulator, trading-off the cross-embodiment

flexibility offered by classical planners that are agnostic to a robot's morphology.

We identify two fundamental problems that have deterred learning cross-embodiment motion planning. First, different manipulators have varying kinematic properties such as link lengths and joint limits, as well as diverse morphologies characterized by their degrees of freedom. Each manipulator operates within a particular configuration-subspace bounded by its joint limits. Thus, training a single neural policy to generate actions spanning multiple bounded subspaces renders cross-embodiment motion planning a challenging task to learn. Second, data for training cross-embodiment policies is difficult to gather as there are only a limited number of embodiments available commercially, which do not fully capture the distribution of possible kinematic variations.

To address the above challenges, we present **Cross-Embodiment Motion Policy (XMoP)**, a framework of data-driven methods to learn neural policies for cross-embodiment motion planning. Our contributions are outlined as follows:

- Our novel control policy uses the robot's physical description, i.e., URDF [30] to operate on link-wise SE(3) observations and plans for task-space end-effector targets across a distribution of 6 and 7 DoF manipulators, enabling *zero-shot* configuration-space plan generalization.
- We propose a 3D semantic segmentation-based model for perceptual cross-embodiment collision detection that shows a $98\%$ recall and *zero-shot* transfers to real-world unseen environments.
- Finally, we combine the control policy with the collision model under a model-predictive framework, achieving $70\%$ success rate for motion planning with unseen robotic manipulators.

To the best of our knowledge, XMoP is the first neural policy for configuration-space planning that *zero-shot* transfers to unseen robotic manipulators. We demonstrate sim-to-real transfer on Franka FR3 and Sawyer robots solving novel planning problems in unstructured real-world environments.

## II. Methodology

### A. Whole-Body Control Formulation

Prior methods for neural motion planning directly predict configuration-space actions that do not generalize across embodiments [22, 12, 14, 33, 5, 9]. We formulate XMoP as a Markovian motion dynamics model $f(p_{t+1:t+H}|p_t, g_t)$ that

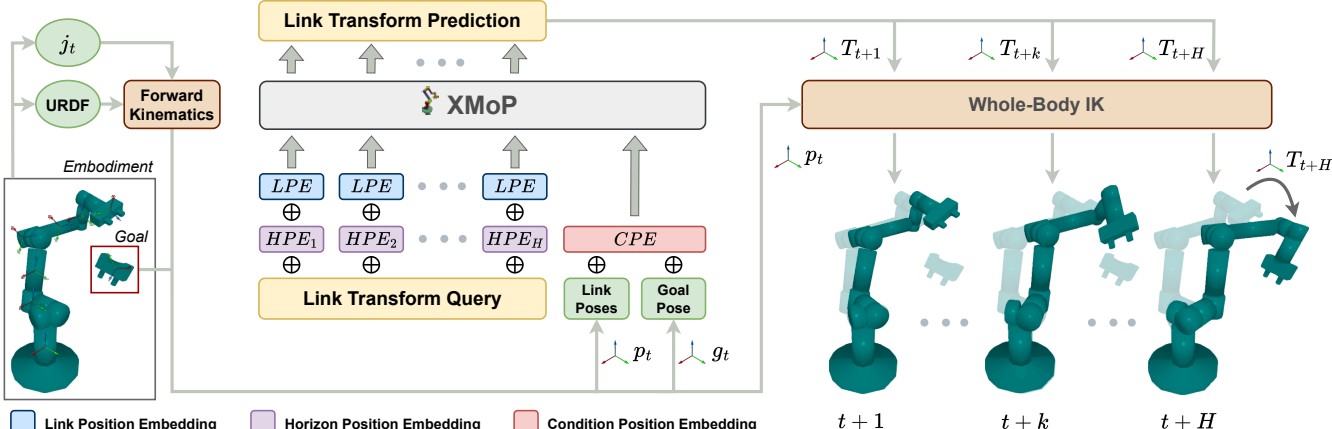

**Fig. 1:** XMoP perceives the embodiment state as a sequence of whole-body SE(3) link poses $p_t$ and predicts link-wise pose transformations $T_{t+1:t+H}$ over a horizon $H$ to move the end-effector towards the goal pose $g_t$. We use a Transformer [31] base policy architecture, that operates on an input pose sequence $(p_t, g_t)$ and uses self-attention to convert the query tokens into a sequence of link-wise relative pose transformations. Contextual information is provided to the Transformer using three types of position embeddings: (1) LPE is a fixed set of sinusoidal position embeddings that are repeated over the horizon providing kinematic chain awareness for every horizon step; (2) HPE and (3) CPE, are learned position embeddings providing awareness for horizon and input, respectively. Additionally, we use novel attention masking strategies within the Transformer for cross-embodiment adaptation. The predicted link-wise transformations are multiplied with the instantaneous link poses to reconstruct the future whole-body pose of the manipulator, which is reached by using a whole-body IK procedure to retrieve the configuration-space joint positions.

provides the future states of manipulator over a horizon of $H$ steps. The instantaneous state observation for the manipulator is represented as a sequence of rigid-body SE(3) link poses with respect to the robot's base i.e., $p_t \in \mathbb{R}^{D \times 4 \times 4}$, where $D$ is the manipulator's degrees of freedom. The goal $g_t \in \mathbb{R}^{4 \times 4}$ represents the end-effector SE(3) target for motion planning. Our policy $\pi(a_t|p_t, g_t)$ as shown in Fig. 1, learns to predict link-wise relative SE(3) transformations $a_t = T_{t+1:t+H}$ for reconstructing the whole-body manipulator pose in future time steps. The formulation for the transformation target $T_{t+k} \in \mathbb{R}^{D \times 4 \times 4}$ for $k \in \{1, ..., H\}$ is shown in eq. 1.

$$T_{t+k}p_t = p_{t+k} \implies T_{t+k} = p_{t+k}(p_t)^{-1} \quad (1)$$

where link poses $p_t$ are obtained using the manipulator's forward kinematics function $\phi(j_t)$, with $j_t \in \mathbb{R}^D$ as the instantaneous configuration-space observation. The configuration-space action in future time step $j_{t+k}$ is retrieved from the predicted whole-body pose $\hat{p}_{t+k} = \hat{T}_{t+k}p_t$ by solving for whole-body IK using the following constrained optimization procedure:

$$\min_{j_{t+k}} \quad \|\hat{p}_{t+k} - \phi(j_{t+k})\|, \quad \text{s.t.} \quad j_L < j_{t+k} < j_U \quad (2)$$

where $j_L$ and $j_U$ are the lower and upper joint limits of the manipulator. The above optimization objective is non-convex, and hence a close initial guess is required for convergence. We address this issue by collecting dense planning demonstrations with a maximum per-joint deviation of 0.05 rad. Thus, making the instantaneous observation $j_t$ to be an initial guess that lies within the close neighbourhood of $j_{t+k}$. In practice, we employ additional stochasticity to handle redundancy and singularities in manipulators.

### B. Pose Transformation Policy

We formulate the whole-body transformation policy $\pi_\theta(a_t|p_t, g_t)$ as a stochastic diffusion policy [7] parameterized by $\theta$ that predicts a batch of possible future trajectories for model predictive control. While training, the noise prediction model $\epsilon_\theta$ takes the noisy sample $a_t^\tau$, which is obtained by applying the forward diffusion process to $a_t^0 = T_{t+1:t+H}$, where $\tau$ is the diffusion step. Additionally, we also pass $c = (p_t, g_t)$ for observation and goal conditioning. For step conditioning, we follow the adaptive layer normalization strategy proposed in Diffusion Transformers [20]. We train the noise prediction model using mean square error loss as shown in eq. 3 which minimizes the variational lower bound of the KL-divergence between the original data distribution $p(a_t^0)$ and the DDPM [11] $q(a_t^0)$ distribution.

$$\mathcal{L}_{xmop-m} = \|\epsilon_\theta(a_t^\tau, c, \tau) - \epsilon\|_2^2, \quad \epsilon \in \mathcal{N}(0, I) \quad (3)$$

Our policy utilizes the Transformer [31] model as the underlying backbone which expects input in a sequence format. We emphasize on four key design decisions in our policy:

1) **SE(3) Proprioception**: The embodiment state is provided to and queried from the policy as a sequence of SE(3) *pose-tokens*, allowing the policy to learn motion synergies between rigid-body links. We use the noisy sample $a_t^\tau$ as transform query, which is passed to the Transformer as *pose-tokens*, as shown in Fig. 1.
2) **Kinematic Masking**: We introduce an inductive bias for kinematics by restricting attention to parent or ancestor links at the current horizon step, and to the same link at both the current and previous horizon steps.
3) **Morphology Adaptation**: To enable learning over different morphologies, we fix the *pose-token* for the end-

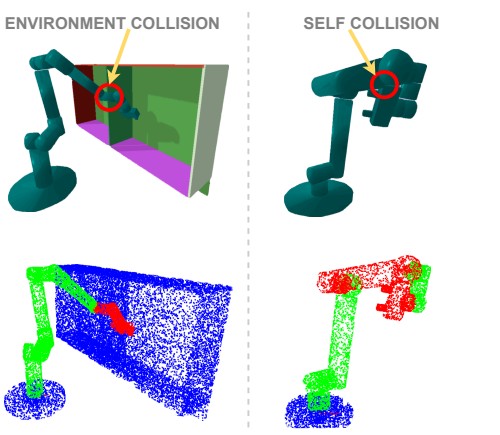

ENVIRONMENT COLLISION     SELF COLLISION

**Fig. 2:** Point-wise training labels for XCoD. ● Collision, ● Not Collision, ● Obstacles.

effector and mask out the *pose-tokens* of unavailable links.

4) **Link-Horizon Position Embedding**: Contextual information is provided to the Transformer using position embeddings for query and input *pose-tokens*. Fig. 1 shows the position embedding scheme used in XMoP.

Prior works have used link-wise tokens [10], temporal state inputs [13, 25], action diffusion [7, 13, 5], and SE(3) pose observations [19, 27] for learning and planning applications. We combine these ideas with our whole-body pose transformation method to learn neural policy for cross-embodiment motion planning.

### C. Collision-Free Motion Synthesis

We formulate collision detection as a semantic segmentation problem for identifying the links of the robot that are in collision given a pointcloud observation of the workspace. Our semantic collision detection model **XCoD** : $\mathbb{R}^4 \to \mathbb{R}^2$ takes segmented pointcloud of the workspace as input where each point consists of 3D spatial coordinates, and a semantic label. These points are uniformly sampled from the surface of individual links of the manipulator and scene obstacles for the future time steps as predicted by our planning policy $\pi_\theta$. We assign unique semantic labels to every link corresponding to its degree of freedom, while using a reserved label for all obstacles. For training the collision model, we utilize point-wise binary label $y$, where collision-free link points are assigned a training label of 0, whereas link points in collision are assigned a training label of 1. Fig. 2 shows planning scenes and corresponding colorized pointclouds highlighting point-wise training labels. We utilize a Point Transformer V3 [32] model for the semantic segmentation task and train it using cross-entropy loss along with an additional surrogate Lovász hinge loss, that has shown to improve semantic segmentation performance in prior works [4].

We use XCoD to assign scores for a batch of trajectories predicted by XMoP and choose the least collision future trajectory for locally reactive planning. eq. 4 shows the formulation of the proposed Model Predictive Control (MPC) method where $N$ is the number of surface points sampled from the

manipulator, $B$ is the prediction batch size, and $\hat{y}$ is the per-point collision logit from XCoD.

$$a_t^* = a_t^{(q)}, \quad q = \arg\min[s_1, s_2, \dots, s_B],$$
$$s = \frac{1}{HN}\sum_{h=1}^{H}\sum_{i=1}^{N}\arg\max \hat{y}_i \tag{4}$$

Similar to diffusion Policy [7], we predict for $H_p$ steps while executing only $H_a$ steps on the manipulator. XMoP avoids geometric biases from manipulator design, by conditioning on task-space observations (SE(3) poses and pointclouds) that implicitly promote cross-embodiment and sim-to-real generalization.

### D. Data Generation and Training

**Kinematic Templates**. Synthetic manipulators are represented with open kinematic chains connecting a series of rigid-body links. We design these links using axis-aligned cylinders forming a rigid-body template. Each link template is parameterized with the following information: (1) length of the cylinders, (2) radius of the cylinders, and (3) constraints for the joint that connects the link to the preceding link. We follow the design pattern of two commercially available robots: (1) 6-DoF UR [2] (2) 7-DoF Sawyer [1]. Fig. 1 (*left*) shows a composed manipulator sampled from our synthetic embodiment distribution by randomizing the parameters for constituent link templates. We adopt the 3.27 million synthetic planning problems from the MπNets dataset [9] and generate demonstration data by sampling a unique embodiment for each problem and solving it using the AIT* [28] motion planner.

**Data Augmentation**. During training, we randomize the position and orientation of the link frames for pose computation by uniformly choosing cylinders from the constituent kinematic templates. With this frame augmentation, the number of possible sequences for a single manipulator is $3^{DoF}$ which promotes cross-embodiment generalization during training.

### III. EXPERIMENTS AND RESULTS

We evaluate XMoP on 7 different robotic manipulators from 5 commercial manufacturers: Franka Panda, Rethink Sawyer, Kuka IIWA, Kinova Gen3 6-DoF, Kinova Gen3 7-DoF, Universal Robots UR5, and Universal Robots UR10. For each manipulator we use a set of 500 novel problems from the MπNets [9] test distribution, ensuring that valid collision-free IK solutions exist for both start and goal end-effector targets. It should be noted that none of the robots in our benchmark experiments or real-world demos were part of the training dataset. All of our results are *zero-shot* evaluations with single pre-trained checkpoints.

### A. Benchmark Evaluations

We evaluate XMoP on benchmark problems for each commercial manipulator. Policy rollouts are terminated if the manipulator's end-effector reaches the goal or a maximum of 200 rollout steps are exhausted. We consider a goal to be reached when the $L2$ norm of the SE(3) pose difference

| Embodiment | XMoP+XCoD | | | AIT*+XCoD | | | AIT*+PyBullet | | |
|---|---|---|---|---|---|---|---|---|---|
| | SR[%] ↑ | PL ↓ | ST[s] ↓ | SR[%] ↑ | PL ↓ | ST[s] ↓ | SR[%] ↑ | PL ↓ | ST[s] ↓ |
| Panda | 71.8 | $4.6 \pm 4.7$ | $49.8 \pm 65.8$ | 86.0 | $3.6 \pm 2.3$ | $39.7 \pm 27.3$ | 94.4 | $2.9 \pm 1.5$ | $4.0 \pm 0.3$ |
| Sawyer | 70.8 | $4.8 \pm 5.3$ | $42.9 \pm 53.6$ | 90.4 | $3.3 \pm 2.8$ | $34.6 \pm 27.1$ | 92.4 | $1.9 \pm 0.9$ | $3.9 \pm 0.5$ |
| IIWA | 71.0 | $5.1 \pm 5.6$ | $38.3 \pm 52.5$ | 87.6 | $2.8 \pm 2.1$ | $32.3 \pm 21.2$ | 93.4 | $2.1 \pm 1.0$ | $3.9 \pm 0.4$ |
| Gen3 6-DoF | 67.6 | $4.7 \pm 6.0$ | $51.8 \pm 70.9$ | 71.0 | $2.5 \pm 2.6$ | $24.2 \pm 11.0$ | 92.4 | $2.0 \pm 0.9$ | $3.9 \pm 0.5$ |
| Gen3 7-DoF | 78.2 | $5.5 \pm 5.9$ | $44.0 \pm 53.0$ | 88.4 | $3.3 \pm 2.6$ | $35.0 \pm 22.6$ | 94.2 | $2.2 \pm 1.2$ | $3.9 \pm 0.4$ |
| UR5 | 70.8 | $3.1 \pm 3.3$ | $42.2 \pm 71.2$ | 80.8 | $2.6 \pm 1.9$ | $31.0 \pm 20.7$ | 88.8 | $2.1 \pm 1.5$ | $3.9 \pm 0.4$ |
| UR10 | 67.4 | $3.1 \pm 3.4$ | $31.5 \pm 52.0$ | 72.6 | $2.9 \pm 2.6$ | $33.2 \pm 24.8$ | 92.2 | $2.1 \pm 1.2$ | $3.8 \pm 0.6$ |

**TABLE I:** Results from our benchmark evaluations. We expect the baseline AIT*+PyBullet to perform better as it has access to the ground truth obstacle information whereas XMoP relies purely on visual inputs.

between end-effector and goal is less than a pre-specified threshold of 0.01.

**Baseline Planners** We compare our policy against the upper performance threshold of AIT* [28] planner that has access to an oracle collision checker from PyBullet. We know that our performance will be worse than AIT* which has access to oracle environment state, however it is the best option available to us for cross-embodiment evaluation as there is no neural motion planner baseline that works across *unseen robot embodiments*. Thus, we do not have a neural motion planning baseline in this work. We also evaluate generalization capabilities of our learned collision model XCoD by combining it with the AIT* baseline. This hybrid planner utilizes the XCoD model for collision queries, where a state is considered invalid if the ratio of detected link points in collision to the total number of manipulator points is greater than 0.001.

We utilize the following quantitative metrics to evaluate the planning performance: (1) *Success Rate (SR)* - A trajectory is successful if the final end-effector position is within 1 cm and orientation is within $5°$ of the goal, with no collisions or joint limit violations. (2) *Path Length (PL)*: Sum of $L2$ norm between consecutive configuration-space way points. (3) *Solution Time (ST)*: Total time elapsed to generate a successful trajectory. Table I shows the benchmark results.

### B. Result Analysis

**Plan Optimality**. Table I shows that the oracle AIT* baseline generates approximately $50\%$ more optimal plans and is 10 times faster compared to XMoP and XCoD MPC policy. However, this comes with the assumption of privileged obstacle information being available to PyBullet for collision detection, thus making it infeasible for deployment in unstructured real-world environments.

***Zero-shot* Generalization**. Table I demonstrates XMoP's ability to generalize to manipulators with novel designs that were unseen during training, as our training data includes only the design patterns of Sawyer and UR robots. XMoP exploits the fact that motion behavior is characterized by the whole-body pose of the embodiment [3, 6]. For similar whole-body poses, different manipulators might have contrasting

joint configurations, but the link poses are relatively closer in SE(3). Similarly, configuration-space actions for different manipulators are dependent on their morphology, but link pose transformations are similar for individual manipulator links.

**Kinematic Constraints Satisfaction**. Our policy predicts link-wise transformations that obey different kinematics constraints across manipulators from a single pose sequence observation. We will investigate these properties of XMoP in future work.

**Collision Detection**. As shown in Table I, the hybrid planner achieves an average $82.4\%$ success rate, which is within $10\%$ of the oracle baseline, thus highlighting the effectiveness of our collision detection model. Although XCoD is trained on complete scene point clouds in simulation, it shows strong generalization to unseen partial point clouds captured from depth cameras in the real world.

**Sim-to-Real Experiments**. We deployed XMoP on two commercial manipulators, Franka FR3 and Sawyer, in different unseen real-world environments. Appendix Fig. 3 shows intermediate snapshots of our policy rollouts. We used mono-color obstacles and segmented them from a calibrated depth camera to extract the obstacle point cloud.

## IV. CONCLUSION

**Limitation**. As with any behavior cloning algorithm, our policy performance is limited by the quality of the synthetic demonstration data available for training and struggles to plan for out-of-distribution goal poses and environment setups.

**Conclusion**. In this paper we presented XMoP, a novel configuration-space neural motion policy for re-targeting planning behavior *zero-shot* to unseen robotic manipulators. We formulated motion planning as a link-wise SE(3) pose transformation method, showcasing its scalability for data-driven policy learning. We used fully synthetic data for training models for motion planning and collision detection demonstrating strong sim-to-real generalization to real robotic manipulators. XMoP is trained using a behavior cloning approach and is capable of planning the motion for a distribution of robots, thereby unlocking a class of neural policies for learning cross-embodiment behaviors.

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

APPENDIX

**XMoP PLANNING ROLLOUT**

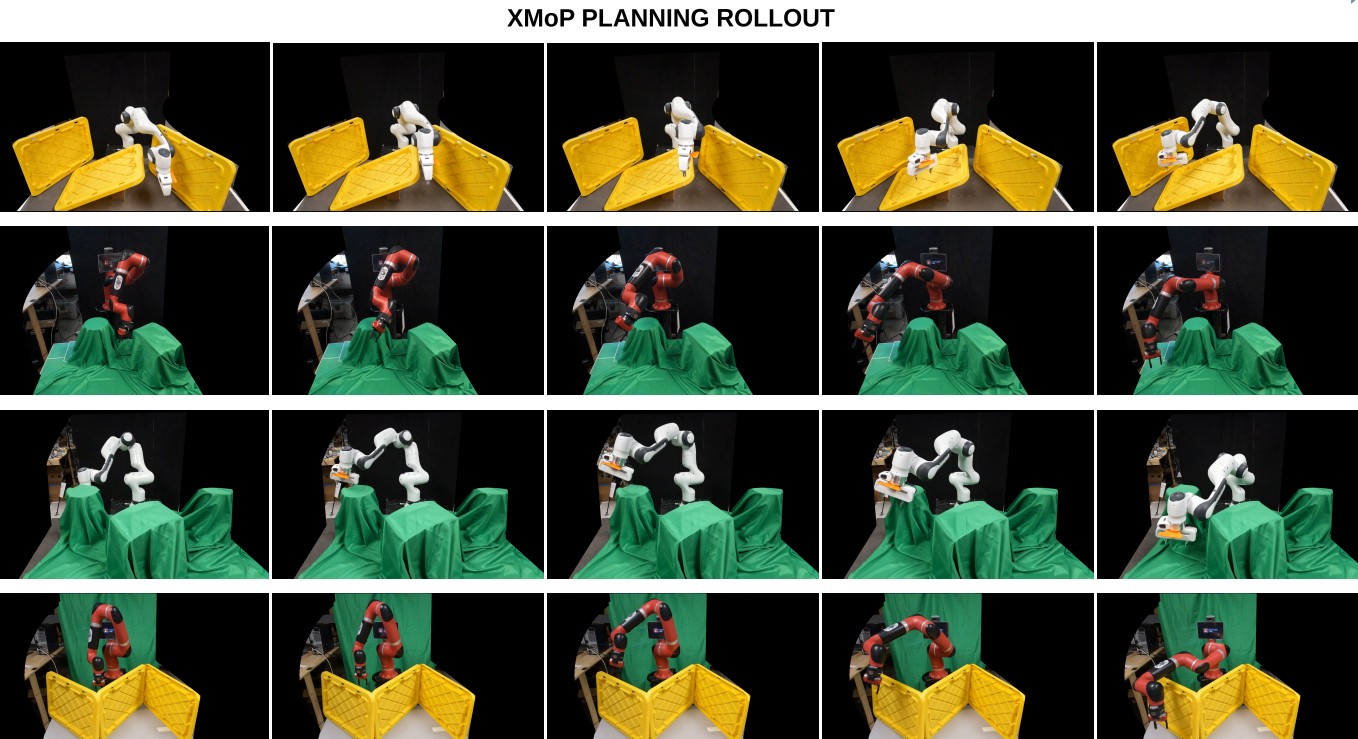

**Fig. 3:** XMoP plans across different unstructured environments for two unseen 7-DoF commercial manipulators Franka FR3 and Sawyer (better viewed when zoomed in). Our benchmark experiments and real-world demos use the same set of frozen policy parameters showcasing *zero-shot* sim-to-real generalization for neural motion planning. Videos of policy rollouts are available at https://sites.google.com/view/xmop.