# OpenReview forum: "Whole-Body Neural Policy for Zero-Shot Cross-Embodiment Motion Planning"
_roboticsfoundation.org/RSS/2024/Workshop/EARL — EARL 2024 Poster_

### Official Review · Reviewer_V4Xv · 2024-06-18

**Rating:** 6
**Confidence:** 5

**Review:**

This paper proposes a solution to a very interesting topic of learning how to plan for a range of robot embodiment. The presented approach is interesting, especially the way of the collision assessment. However, the way of handling the kinematic chain constraints is not well described and the numerical results are quite far from being convincing.

Pros:
- interesting approach to handle different robot embodiment
- the way of detecting collisions that works on the raw data


Cons:
- the description of handling the kinematic chain constraints is not clear enough
- the obtained results suggests that the proposed method introduces quite a big computational overhead over the classical planning approaches, which makes the use of learning questionable (typically we use learning to minimize the plan computation time)

Comments and questions:
1. "Furthermore, these algorithms conventionally assume the availability of a precomputed geometric representation of the robot’s workspace
for state validation, thus making them unusable in unseen
environments with novel types of obstacle" - I agree that typically planers require a map, but it is not making classical motion planning approaches unusable if a "novel type" of obstacle is present. Maybe this point refers to some trajectory optimization techniques, but this is not true for a wide range of planners.

2. How the kinematic chain constraints are enforced? Are they just ignored and resolved by the IK, which minimizes the error but never reaches values close to 0? Authors wrote "Our policy predicts link-wise transformations that obey different kinematics constraints across manipulators from a single pose sequence observation" but it is very unclear why and to what extent this is true.

3. Authors emphasize that this planner works in unseen environments, however, it is not clear what is meant by that. Reviewer understands that this refers to the fact that there is no map of the environment accessible. However, authors seems to assume that they have access to complete point clouds of the environment, which can be easily transformed to a map. This creates a confusion.

4. This is especially concerning taking into account the results, which kind of shows that if we can generate a map of the environment in less than 40s then AIT* is more efficient than the proposed planner.

5. The description of the proposed data augmentation technique is hard to understand.
From the number of combinations it seems that for every joint there are 3 choices, however, it is not clear how this emerges from "uniformly choosing link templates". It is also not clear how choosing cylinders randomly allows for satysfying the constraints of the kinematic chain.

6. At one place authors wrote " We consider a goal to
be reached when the L2 norm of the SE(3) pose difference
between end-effector and goal is less than a pre-specified
threshold of 0.01" while in the other place "A trajectory
is successful if the final end-effector position is within 1 cm
and orientation is within 5◦ of the goal, with no collisions
or joint limit violations" which are not equivalent.

7. Authors wrote "XCoD by virtue of its formulation
learns a robust 3D mesh collision classifier conditioned on
link-wise semantic labels specified for the robot" but from the introduction of XCoD it seems that the collision classification is done based on point clouds not meshes.

8. Limitation one can be addressed by learning-to-plan works that does not consider the need for an oracle [1,2] or can improve over the plans generated by an oracle [3].

[1] T. Jurgenson, A. Tamar, "Harnessing Reinforcement Learning for Neural Motion Planning", in Proceedings of Robotics: Science and Systems, 2019, Freiburg im Breisgau, Germany, doi: 10.15607/RSS.2019.XV.026

[2] P. Kicki et al., "Fast Kinodynamic Planning on the Constraint Manifold With Deep Neural Networks," in IEEE Transactions on Robotics, vol. 40, pp. 277-297, 2024, doi: 10.1109/TRO.2023.3326922.

[3] P. Kicki and P. Skrzypczyński, "Speeding up deep neural network-based planning of local car maneuvers via efficient B-spline path construction," 2022 International Conference on Robotics and Automation (ICRA), Philadelphia, PA, USA, 2022, pp. 4422-4428, doi: 10.1109/ICRA46639.2022.9812313

9. " k ∈ {1...H}" missing commas

10. Some of the references are to ArXiv while they are already published works.

---

### Decision · Program_Chairs · 2024-06-24

Accept (Poster)